behaviour/cognition/neuroscience

hand representation, adaptation, anthropoid hand, visual processing, occipitotemporal cortex

**Author for correspondence:**
Massimiliano Conson
e-mail: massimiliano.conson@unicampania.it

# 'Not only faces': specialized visual representation of human hands revealed by adaptation

Massimiliano Conson[1], Francesco Polito[1], Alessandro Di Rosa[1], Luigi Trojano[1], Gennaro Cordasco[1], Anna Esposito[1] and Marco Turi[2]

[1]Department of Psychology, University of Campania Luigi Vanvitelli, Caserta, Italy
[2]Stella Maris Mediterraneo Foundation, Chiaromonte, Potenza, Italy

(iD) MC, 0000-0002-4776-306X

Classical neurophysiological studies demonstrated that the monkey brain is equipped with neurons selectively representing the visual shape of the primate hand. Neuroimaging in humans provided data suggesting that a similar representation can be found in humans. Here, we investigated the selectivity of hand representation in humans by means of the visual adaptation technique. Results showed that participants' judgement of human-likeness of a visual probe representing a human hand was specifically reduced by a visual adaptation procedure when using a human hand adaptor but not when using an anthropoid robotic hand or a non-primate animal paw adaptor. Instead, human-likeness of the anthropoid robotic hand was affected by both human and robotic adaptors. No effect was found when using a non-primate animal paw as adaptor or probe. These results support the existence of specific neural mechanisms encoding human hand in the human's visual system.

## 1. Introduction

Primates, and in particular humans, are remarkably able to navigate the social environment by relying upon an exceptional ability to process fundamental visual signals about other persons' face, as facial expression and eye gaze, as well as about other persons' body, as bodily posture and movement [1,2].

The striking ability to deal with these social signals suggested that the primate brain is equipped with neural systems specialized for processing socially relevant information [2]. The most compelling evidence comes from neurophysiological studies revealing single neurons in the monkey inferotemporal cortex selectively discharging

to the sight of faces and of eye gaze [3,4]. Moreover, classical neurophysiological studies also revealed neurons in monkey inferior temporal cortex selectively discharging to the vision of the hand; in particular, these neurons discharged specifically to the presentation of monkey and human hands rather than to the presentation of shapes different from human hands [5,6].

In humans, neuroimaging studies [7] and electrophysiological recording from the cortical surface [8] demonstrated the specific involvement of the occipitotemporal cortex in face processing, and psychophysical experiments confirmed the existence of neural mechanisms selectively encoding faces [9]. Both neuroimaging and event-related potentials studies also revealed differential responses in the extrastriate visual cortex to the visual presentation of hands and of other body parts [10–12]. Consistently, one electrophysiological study comparing static images of hands, faces, cars or flowers showed hand-specific responses in the cortical surface of both superior temporal and parietal regions [13].

A well-recognized method to test the existence in humans of neural populations selectively encoding specific stimulus categories is adaptation. Indeed, psychophysical adaptation experiments are used to infer the tuning properties of cells underlying the perception of specific stimuli, since the responses of neurons tuned to the adapting stimulus are selectively reduced by repeated exposure [14]. In particular, in the visual domain, adaptation is the loss of responsivity in cells coding a precise stimulus feature due to the prolonged exposure to that particular feature. After adaptation, the perceptual judgement of that feature is reduced, and the strength of this after-effect depends on the similarity between the adaptor and the test stimulus [14].

Adaptation has been extensively used to uncover neural mechanisms selectively encoding both low-level perceptual features, such as motion direction [15] and orientation [16], and high-level stimuli, such as faces [9] and goal-directed hand actions [17].

In a study on face adaptation, Kovàcs et al. [18] demonstrated that exposure to female faces produced a significant face's gender after-effect since probe faces were perceived as more masculine as compared with the control condition. Interestingly, moreover, the authors found that adaptation to faces did not affect the gender judgements of human hand probe stimuli, and vice versa. Since Kovàcs et al. [18] also found that hand's gender after-effects were not affected by changes in hand size or orientation, they suggested that this gender after-effect occurred from the adaptation of the high-level shape-specific mechanisms of faces and hands.

Hence, the mechanisms of human perception revealed by visual adaptation paradigms show remarkable parallels to the neural mechanisms revealed in monkey by neurophysiological techniques [14,15]. In the present study, we used visual adaptation to investigate in humans the specificity of neural mechanisms coding the visual representation of the human hand [10–12]. In particular, we tested cross-category adaptation [19] by comparing three stimulus categories and investigated whether human hand, robotic anthropoid hand and non-anthropoid animal paw used as adaptors could affect human-likeness judgement on probe stimuli of the same three categories. Evidence of a neural representation specifically encoding the human hand [11–13] would be supported by results demonstrating that participants' human-likeness judgement of a human hand probe is specifically affected by the adaptation procedure derived by using as adaptor a human hand but not when the adaptor is a robot or a non-anthropoid animal hand.

# 2. Material and methods

## 2.1. Participants

A sample of 49 university students (all males; mean age = 25.5 years, s.d. = 3.1) recruited at the Developmental Neuropsychology Laboratory (Department of Psychology, University of Campania 'Luigi Vanvitelli') volunteered to participate to the present study. All the participants were right-handed males, neurologically healthy, without psychiatric or other medical disorders.

The entire protocol was approved by the Local Ethics Committee of the Department of Psychology, University of Campania Luigi Vanvitelli and was conducted in accordance with the ethical standards of the Helsinki declaration; written informed consent was obtained from all participants before the experiments.

## 2.2. Stimuli

The experimental stimuli were grey-scale photos of human male hands, robotic hands and paws, each stimulus category including two different identities portrayed from the same perspective: from dorsum and with fingers pointing up (figure 1a).

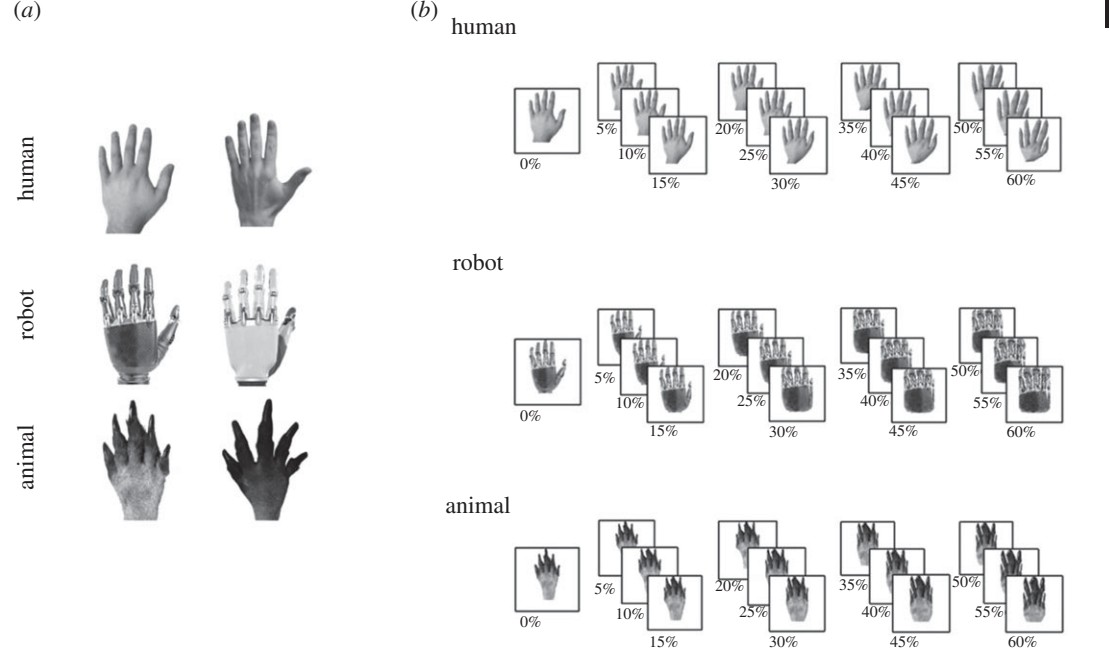

**Figure 1.** (*a*) Experimental stimuli (on the left the item #1 and on the right the item #2 for each of the three categories) before applying the morphing procedure. (*b*) Examples of stimuli (one for each of the three categories) morphed from 0% to 60%.

All images were acquired in full colour, then were converted to grey-scale and matched for contrast and brightness by using Adobe Photoshop CS5 software. A morphing procedure was thus applied through the 'scale based on content' function of Adobe Photoshop CS5 software, modifying each image on both the horizontal and the vertical dimensions and varying the value of the 5% function at regular intervals from 0% to 60%. By this means, we could manipulate the human-likeness of the images by modifying the configural shape of the hand (or paw) while leaving the focus of the process unchanged for all images (figure 1*b*). To verify whether human-likeness of the images was actually manipulated and whether human-likeness of two items within each category was comparable and differed from human-likeness of each item of the other categories, we asked 30 healthy male volunteers (not recruited for the main experiment; age range: 19–27 years) to rate the degree to which each stimulus was similar to a human hand (human-likeness) on a 1-to-10 Likert scale [20]. Mean human-likeness judgement at 0% morphing differed between the three categories, whereas it did not differ between the two identities within each category (hand #1: mean = 9.3, s.d. = 1; hand #2: mean = 8.7, s.d. = 2.3; robot #1: mean = 7.1, s.d. = 2; robot #2: mean = 6.7, s.d. = 2.6; paw #1: mean = 2.6, s.d. = 2.1; paw #2: mean = 2.1, s.d. = 1.4). Indeed, the results of a univariate ANOVA, with item as independent variable, demonstrated a main effect of the item, $F_{5,174} = 70\,592$, $p = 0.0001$, $\eta_p^2 = 0.67$. Bonferroni-corrected *post hoc* comparisons did not reveal differences between the two items within each of the three categories (all $p > 0.05$), whereas each item within a single category differed from items of the other categories (all $p < 0.003$).

The stimuli subtended a visual angle of approximately 6° × 4° (at a viewing distance of 50 cm from a 19-inch computer monitor), whereas the adaptors were enlarged by 25%, in order to avoid low-level perceptual adaptation [21].

## 2.3. Baseline and adaptation

The experiment comprised three phases: a first baseline phase, an adaptation phase and second baseline phase run in the same session [21] (figure 2).

The first baseline phase comprised two identical blocks (baseline 1 and 2). In each block, the two human hands, the two anthropoid robot hands and the two paws, and their morphed shapes were shown to the participants (6 items × 13 levels of morphing, for a total of 78 items). Each trial consisted of a probe presented for 200 ms. Participants were required to judge for each probe how much it looked like a human hand on a scale from 1 to 10 (human-likeness judgement). Presentation order was randomized. Baseline 1 was used to familiarize participants with the task (practice), and its results were disregarded.

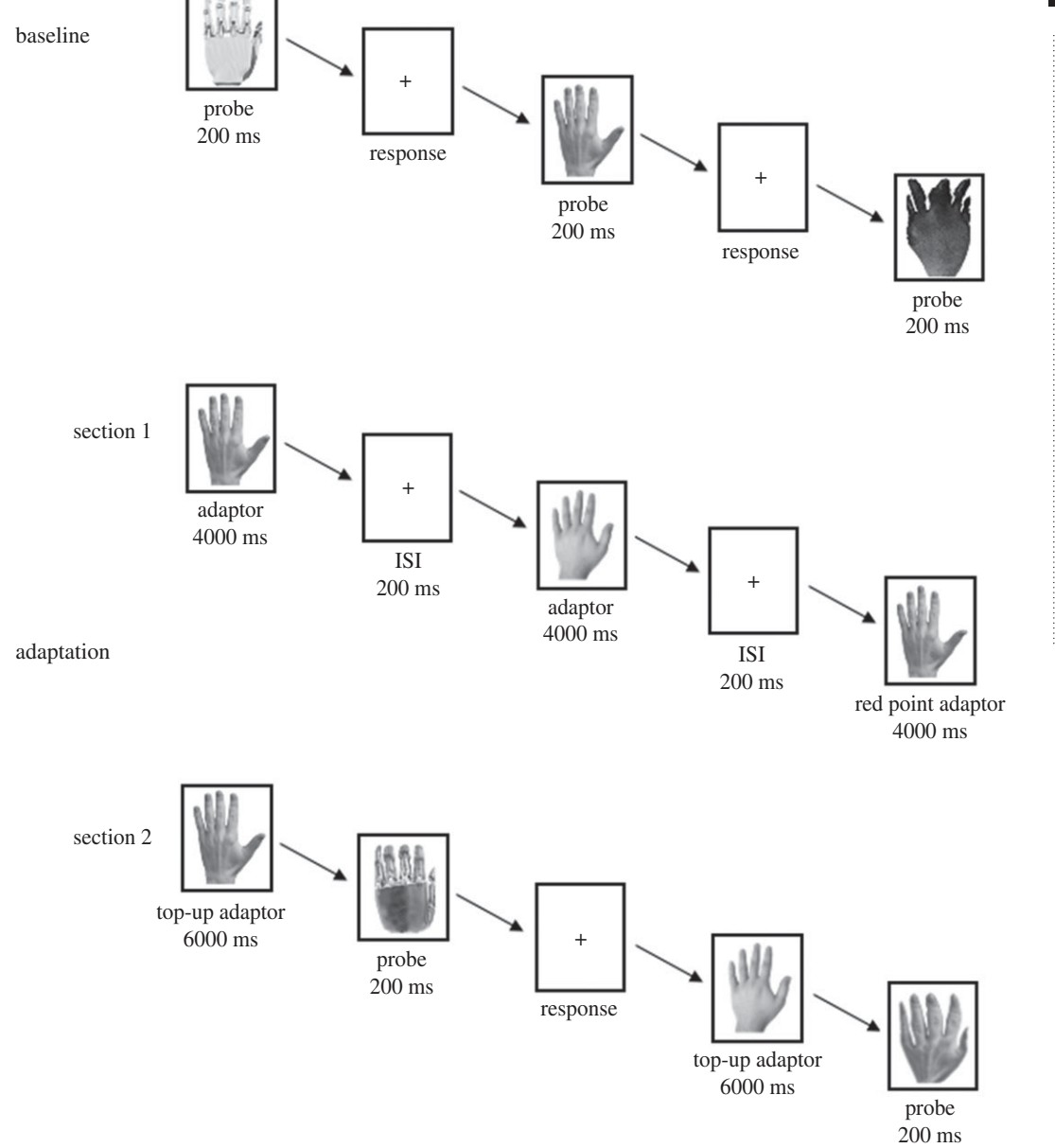

**Figure 2.** Trial format and experimental procedure. The trial format of baseline 1 was the same as that of baseline 2.

The adaptation phase contained two sections. Section 1 included three conditions, one for each adaptor: human hands, anthropoid robot hands and paws; the presentation order of conditions was randomized for each participant. In each condition, two adaptor images were repeatedly presented for 4000 ms each (60 images in total with a 200 ms ISI to eliminate any apparent motion). Some images (8% of the trials) presented red dots and the participants were required to count how many red dots they saw on the adaptor to ensure they paid attention to the stimuli throughout. The total presentation time of the adaptor was of 4 min.

After completing each condition of section 1, the participants underwent section 2 of the adaptation phase containing the same probe stimuli as the baseline blocks. Again, the participants were required to judge the human-likeness of the probes. However, each probe was preceded by a 6000 ms top-up adaptor, to maintain the adaptation effect. The type of top-up adaptor (human hand, anthropoid robot hand or paw) depended on type of the condition the participants were performing. The presentation order of the three probe categories (human hands, robot hands and paws) was randomized.

The second baseline phase was identical to the first one, comprising two identical blocks (baseline 3 and 4). Baseline 3 served to dissipate any remaining effects of adaptation and was not included in the analysis. Thus, baselines 2 and 4 were used as a measure of the baseline performance.

## 2.4. Statistical analysis

Raw data are available in [22]. Our primary analysis measured adaptation as a change of human-likeness judgement from the mean judgement of the baseline phases (2 and 4), expressed in z-score, in the three adaptor conditions provided on the *human hand probe*. Negative scores mean that subjects are less likely to judge the stimulus after adaptation as human (repulsive effect), whereas positive scores indicate that subjects are more likely to judge it as human (attractive effect). A $3 \times 13$ repeated measures ANOVA was performed on human-likeness judgement of hand probe, with adaptor (hand, robot and paw) and morphing level (from 0 to 60%) as within-subject factors. Then, we tested whether adaptation occurred on robot hand and paw probes. Thus, the same $3 \times 13$ repeated measures ANOVA as above was performed separately on human-likeness judgement of robot hand probe and of paw probe, with adaptor (hand, robot and paw) and morphing level (from 0 to 60%) as within-subject factors.

Greenhouse–Geisser correction was used whenever the assumption of sphericity was violated, but uncorrected degrees of freedom were reported for transparency. Bonferroni-corrected *post hoc* comparisons were performed when necessary.

## 3. Results

Results of the ANOVA on human-likeness judgement of *human hand probe* showed a significant main effect of morphing, $F_{12,576} = 14.60$, $p = 0.0001$, $\eta_p^2 = 0.092$, with the following significant differences: 0% versus 15%–60%, 5% versus 10%–60%; 10% versus 15%–60%; 15% versus 25%; 20% versus 50% and 60%; 25% versus 30%, 35% and 45%–60%; 30% versus 40%; 35% versus 40%: 40% versus 50% and 60%; 45% versus 50% and 60%; 50% versus 55% (all $p < 0.05$).

The main effect of the adaptor showed a trend towards significance, $F_{2,96} = 2.471$, $p = 0.08$, $\eta_p^2 = 0.081$, figure 3a). Instead, the adaptor × morphing interaction was significant, $F_{24,1152} = 6.30$ $p = 0.0001$, $\eta_p^2 = 0.084$; *post hoc* comparisons showed no differences between the three adaptation conditions from 0 to 15% of morphing ($p > 0.05$); from 20 to 35% the score for human hand adaptation was significantly lower with respect to the others two adaptation conditions (all $p < 0.01$), and from 40 to 60% the three adaptation conditions were not statistically different ($p > 0.05$) (figure 3b).

Results of the ANOVA on human-likeness judgement of *robot hand probe* showed a significant a significant main effect of morphing, $F_{12,576} = 18.88$, $p = 0.0001$, $\eta_p^2 = 0.112$, with the following significant differences: 0% versus 15%–60%; 5% versus 15%–60%; 10% versus 15%–60%; 15% versus 20% and 50; 20% versus 50%; 25% versus 50%; 30% versus 50%; 35% versus 50%: 40% versus 45% and 50%; 45% versus 50%; 50% versus 55% (all $p < 0.05$). The main effect of the adaptor was not significant, $F_{2,96} = 1.232$, $p = 0.29$, $\eta_p^2 = 0.017$ (figure 4a). Instead, the adaptor × morphing interaction was significant, $F_{24,1152} = 2.863$, $p = 0.0008$, $\eta_p^2 = 0.048$, with no differences between the three conditions from 0 to 15% of morphing ($p > 0.05$); at 20% of morphing, human hand adaptation differed from paw adaptation condition ($p = 0.001$), and at 25%, 40%, 45% and 50%, the effect of robot hand adaptation differed from the paw condition (all $p < 0.03$) (figure 4b).

Results of the ANOVA on human-likeness judgement of *paw probe* showed a significant main effect of morphing, $F_{12,576} = 4.176$, $p = 0.0001$, $\eta_p^2 = 0.026$, with the following significant differences: 0% versus 30%, 40%, 50%, 55% and 60%; 5% versus 15%, 20%, 30%, 40%, 50%, 55% and 60%; 10% versus 30-to-45% and 60%; 15% versus 30%, 40%, 45% and 60%; 15%-to-25% versus 30%, 45%, 50% and 60%; 30%–35% versus 40%, 45% and 60% (all $p < 0.05$). The main effect of the adaptor, $F_{2,96} = 0.122$, $p = 0.880$, $\eta_p^2 = 0.002$, and the adaptor by morphing interaction, $F_{36,1152} = 1.289$, $p = 0.158$, $\eta_p^2 = 0.018$, were not significant (figure 5a,b).

Box plots displaying individual data for all experimental conditions are reported in electronic supplementary material, figure S1.

## 4. Discussion

Our study supports data showing that the human visual system is equipped with selective neural mechanisms encoding the visual representation of human hand [10–12]. Indeed, the human hand probe looked less like a hand after adaptation to human hand (repulsive effect) but not to anthropoid robotic hand or to non-primate animal paw. Moreover, human-likeness of robotic hand was reduced after both human and robotic hand adaptation, whereas no effect was observed when using non-primate animal paw as both adaptor and probe.

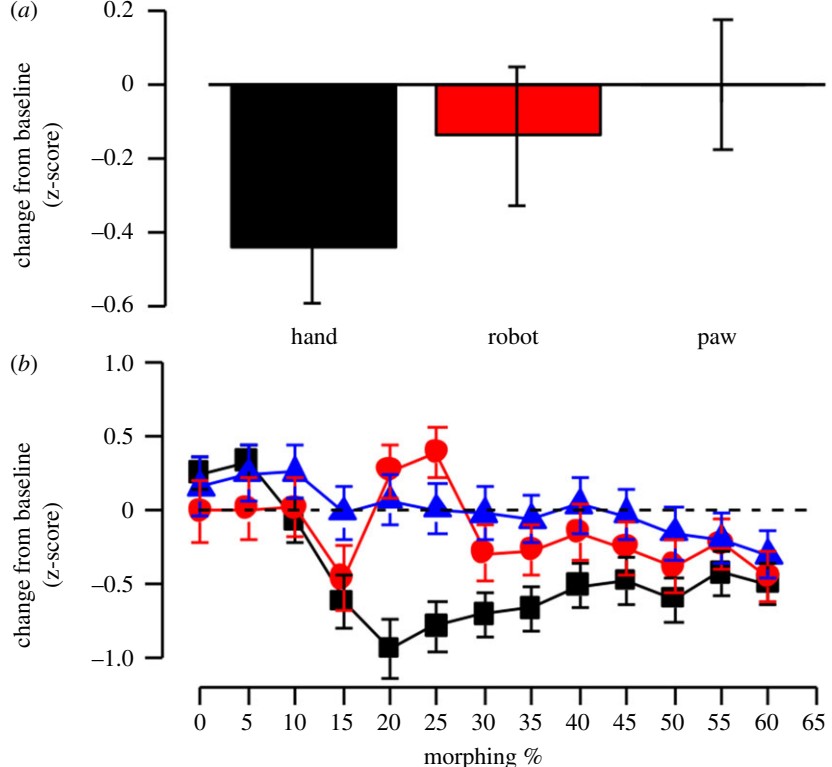

**Figure 3.** (*a*) Mean change from baseline of human-likeness judgement in z-score on human hand probe in the three adaptor conditions. (*b*) Mean change from baseline of human-likeness judgement in z-score on human hand probe in the three adaptor conditions as a function of morphing level of the probe. Error bars correspond to ± 1 s.e.m.

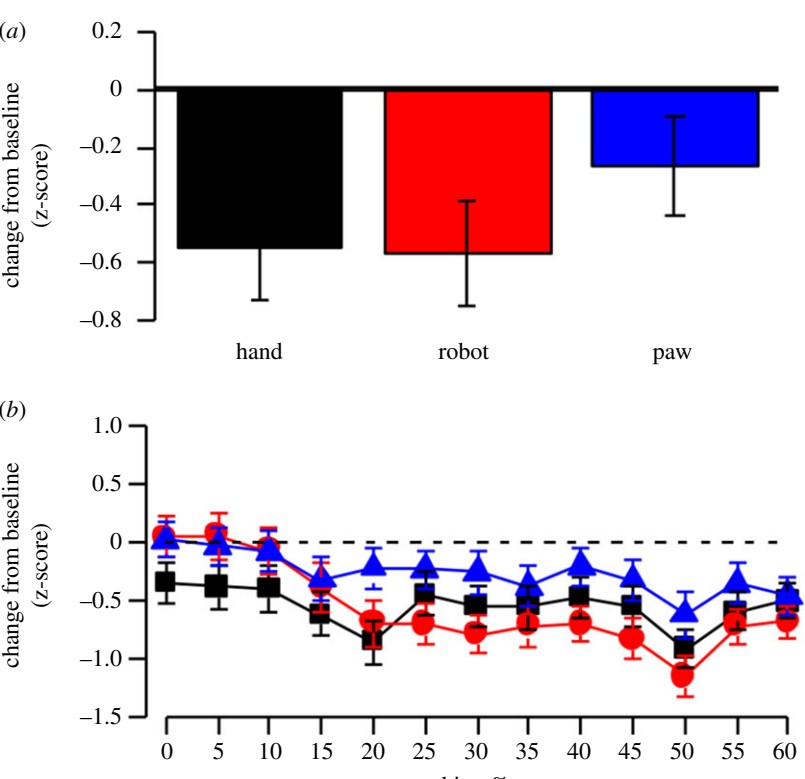

**Figure 4.** (*a*) Mean change from baseline of human-likeness judgement in z-score on robot hand probe in the three adaptor conditions. (*b*) Mean change from baseline of human-likeness judgement in z-score on robot hand probe in the three adaptor conditions as a function of morphing level of the probe. Error bars correspond to ± 1 s.e.m.

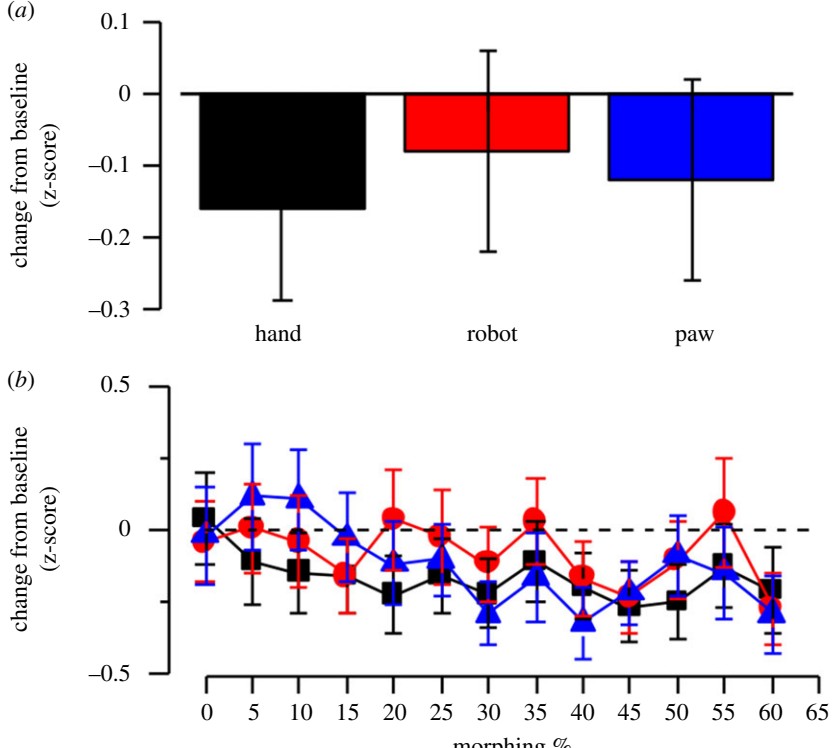

**Figure 5.** (a) Mean change from baseline of human-likeness judgement in z-score on paw probe in the three adaptor conditions. (b) Mean change from baseline of human-likeness judgement in z-score on paw probe in the three adaptor conditions as a function of morphing level of the probe. Error bars correspond to ± 1 s.e.m.

In the hand condition, hand adaptor significantly affected human-likeness of hand probe at levels of morphing representing the centre of the morphing space, i.e. 20–40%. This pattern of adaptation could be explained by the fact that at the boundaries of the morphing range the stimuli were non-ambiguous (human-likeness values were higher at lower morphing levels and, instead, were lower at higher morphing levels), while in the 20–40% range the hand stimuli appeared more ambiguous (human-likeness values were placed in the middle range of the human-likeness scale). This interpretation is consistent with available literature showing that after-effects are minimal for non-ambiguous stimuli, while they are stronger for more ambiguous stimuli [23].

The present findings are consistent with classical neurophysiological data in the monkey brain. Desimone *et al.* [5] investigated the stimulus-selective properties of neurons in the inferior temporal cortex and could identify face-selective cells and also hand-selective cells. The hand-selective neurons responded best to the outline of a variety of monkey and human hands, and their responses dropped or disappeared as the shape of the hand was altered to reduce human-likeness. For instance, a grating-like hand mimicking the periodicity of the fingers and the configuration of spokes radiating from one side of a central core elicited much smaller responses than hand-like shapes. Moreover, the response of the hand-selective cells was enhanced when the hand had a skin colour, appropriate texture and internal details.

The present results suggest that a combination of both functional (fingers/hand morphology) and non-functional (skin colour, texture and internal details) morphological features contribute to define the visual representation of the human hand. Since adaptation to robot hand affected human-likeness judgement of robot hand probe but not of human hand probe, a hierarchical hand representation could be envisaged, with a higher and canonical representation of human hand combining both function and morphology, and a lower human hand representation only defined by the functional specificity of the anthropoid hand, thus leaving out paws. Therefore, a high-level canonical representation of human hand can modify the human-likeness judgement of both canonical and non-canonical anthropoid hand, while a low-level functional anthropoid hand representation (as the robot hand) can affect human-likeness judgement of the robot hand but not of the higher level human representation. The paw does not belong to the hand category thus no cross-category adaptation can be produced.

According to a modular interpretation, specific processes would encode a canonical visual representation of human hand combining shape and other morphological features, such as the skin and the nails, while others would selectively represent the anthropoid hand shape. This interpretation is consistent with neuroimaging in humans showing that the visual representation of body parts in the occipitotemporal cortex is organized based on their functional properties: the hand representation would cluster with the representation of other effectors, such as feet and legs, and separately from facial parts [24]. More interesting here, in human left occipitotemporal cortex two hand-sensitive subregions have been identified with different characteristics: the lateral occipital sulcus, a hand-sensitive region most strongly responding to human hands but also to anthropoid robotic hands, and the extrastriate body area [25], mainly responding to body parts, followed by hands and feet, but not to robotic hands [10]. According to the modular perspective, a hand representation combining functional and morphological (non-functional) aspects of the hand could coexist with an anthropoid hand representation only coding functional hand shape.

It has been suggested that object specific representations in occipitotemporal cortex are built on familiarity and expertise with a given object category [26,27]. This expertise-related view predicts enhanced automatic processing of subordinate levels of a category allowing the observer to automatically and easily recognize the different levels within that given category [26–28]. In this view, a higher experience with human hands would determine specialization of neural structures representing the human hand and, in turn, allowing automatic identification of the low-level category, i.e. the robot hands, but not vice versa. No cross-stimulus effect can involve non-anthropoid animal hands (paws) since paw is not represented as a hand but is rather represented as a different category, namely the paw.

The present study does not allow to clarify the way through which the visual representation of the human hand is built in the human brain and this should be investigated by tailored behavioural paradigms [26,27]. Moreover, it is worth underlining that here we tested adaptation to human male hands in male participants, thus preventing us from taking into account possible effects of sex differences in hand representation [29,30] on the adaptation phenomenon. Notwithstanding these limitations, here we showed that a human visual representation of hand exists [10–12], which combines morphological features related to function (anthropoid hand shape) with morphological features unrelated to function (skin, texture and internal details). Such complex of visual features makes the human hand unique and represents the prototype of the human hand category.

Ethics. The entire protocol was approved by the Local Ethics Committee (Department of Psychology, University of Campania Luigi Vanvitelli, prot. no. 29/2020) and was conducted in accordance with the ethical standards of the Helsinki declaration. Written informed consent was obtained from all the subjects before participating in the experiments.

Data accessibility. Data have been stored in Dryad and are available at https://dx.doi.org/10.5061/dryad.p5hqbzkn4 [22].

Authors' contributions. M.C. was involved in conceptualization, formal analysis, methodology, project administration and writing. F.P. was involved in conceptualization, formal analysis, methodology and writing. A.D.R. was involved in data curation, formal analysis, methodology and project administration. L.T. was involved in methodology, writing and editing. G.C. was involved in resources, software and data curation. A.E. was involved in software and formal analysis. M.T. was involved in conceptualization, formal analysis, writing and editing.

Competing interests. We declare we have no competing interests.

Funding. The research leading to these results has received funding from the EU H2020 research and innovation programme under grant agreement no. 769872 (EMPATHIC) and no. 823907 (MENHIR), the project SIROBOTICS that received funding from Italian MIUR, PNR 2015–2020, D.D. 1735, 13/07/2017, and the project ANDROIDS funded by the programme V:ALERE 2019 Università della Campania Luigi Vanvitelli, D.R. 906 del 4/10/2019, prot. no. 157264,17/10/2019. For author M.T., the research was supported by the EU Horizon 2020 research and innovation programme under grant agreement no. 832813 Spatio-temporal mechanisms of generative perception – GenPercept.

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
