## [Reviewer comments · Royal Society Open Science]

Review History

RSOS-200948.R0 (Original submission)

Review form: Reviewer 1

Is the manuscript scientifically sound in its present form?

Yes

Are the interpretations and conclusions justified by the results?

No

Is the language acceptable?

No

Do you have any ethical concerns with this paper?

No

Have you any concerns about statistical analyses in this paper?

No

Recommendation?

Reject

Comments to the Author(s)

The authors used a visual adaptation behavioural paradigm to test selective mechanisms to the human hand. The results are interesting, but confirm previous evidence, rather than provide novel evidence as, instead, suggested by the authors.

The experiment is sound. The main issue to me is how results are presented. Throughout the paper, the authors repeatedly suggest that their results provide novel evidence for specialized neural mechanisms selective to process the human hand. However, this evidence was long ago reported with neurophysiology in non-human primates (Desimone et al., 1984; Gross et al., 1969) and extensively with neuroimaging in humans (Bracci et al., 2010; 2013; Op de Beeck et al., 2010; Orlov et al., 2010 etc). Interestingly, an fMRI study published 10 years ago (Bracci et al., 2010) already ruled out the role of shape similarity including robotic hands among the control categories.

I would suggest toning down the writing and better acknowledge the already existing extensive evidence on this matter. For instance, the following sentences are not acceptable.

“It is still far from clear, instead, whether humans and monkeys also share specialized brain systems encoding a visual representation of hands”.

“Whether similar representation is found in humans is not known”.

The authors report some of the evidence in the literature, but then go on saying “However, no straightforward data are available on existence in humans of neural populations selectively encoding hand shape.”

The above sentences are not true and also misleading, since the authors are well aware of the existing literature. They cite the evidence but they seem to believe that “good” evidence for neural mechanisms encoding hands can be provided with behavioural data, not neuroimaging data: “Such evidence can be provided by psychophysical studies using visual adaptation”.

I would suggest presenting the results as confirmation of existing literature rather than novel data providing evidence from neural mechanisms not shown before.

Other misleading sentences that point to novelty rather than confirmatory evidence are the following:

“Our study provides evidence that the human visual system is equipped with selective neural mechanisms encoding the visual representation of human hand”.

“These results support the view that neural mechanisms representing faces can be differentiated from those representing hands, but more importantly allow to demonstrate that a shape-selective mechanism specifically coding human hand can be identified”.

Review form: Reviewer 2**Is the manuscript scientifically sound in its present form?**

No

Are the interpretations and conclusions justified by the results?

No

Is the language acceptable?

Yes

Do you have any ethical concerns with this paper?

No

Have you any concerns about statistical analyses in this paper?

No

Recommendation?

Major revision is needed (please make suggestions in comments)

Comments to the Author(s)

The manuscript addresses the question of whether humans have neurons that selectively represent the visual shape of the primate hand, by conducting adaptation experiments using human hands, robot hands and non-human animal paws. The authors conclude that the fact that judgements of 'human-likeness' of a visual probe representing a human hand were reduced when using a human hand adaptor but not a robot or a non-human animal one offers support for the hypothesis that there are specific neural mechanisms encoding human hands.

The stimuli section is lacking in detail: e.g. there is no information about how many different stimuli were created, and it's not clear to me what was actually morphed (I presume items from different stimulus categories were morphed e.g. in Figure 1B, the human hand looks like it's been morphed with an animal paw to me?)

I'm also a little concerned about the stimulus selection. Using only two exemplars for each category seems problematic, especially when they appear to be quite different in terms of low-level features (e.g. the paw stimuli appear to be much darker than the human hand stimuli). I appreciate that you changed the sizes of the adaptors in order to try to counteract this, but it seems to me that if the adaptors were in the same position (just larger) you might still expect low level adaptation that could transfer to the probe stimuli? I am not totally convinced that your results couldn't be explained by the differences in luminance between your stimuli and would appreciate further clarification from the authors on why they think that higher level mechanisms should be evoked.

If I understand the results on P.12 correctly, the only significant interaction effects seem to be for a small range of morphing conditions (20-35%). Some discussion of this pattern would be useful – is it what would be expected based on your stimulus design?

It would also be nice to see individual level data on Figures 3A, 4A and 5A (e.g. in the form of dots superimposed on the bar graphs) in order to get a more intuitive understanding of the variability between participants (which appears quite large, though there currently isn't any indication of what the error bars represent on these graphs).

There are a number of minor typographical errors (e.g. "we showed that participants' judgement of human-likeness of a visual probe representing (a) human hand" in the abstract on P3, incorrect font in P2 on P11, "with no differences between the four three conditions from 0-to-15% of morphing" on P12) and thus I would recommend the authors carry out a full proofread of the manuscript.

Decision letter (RSOS-200948.R0)

Dear Professor Conson

The Editors assigned to your paper RSOS-200948 "NOT ONLY FACES": SPECIALIZED VISUAL REPRESENTATION OF HUMAN HANDS REVEALED BY ADAPTATION" have now received comments from reviewers and would like you to revise the paper in accordance with the reviewer comments and any comments from the Editors. Please note this decision does not guarantee eventual acceptance.

Please submit your revised manuscript and required files (see below) no later than 21 days from today's (ie 03-Sep-2020) date. Note: the ScholarOne system will 'lock' if submission of the revision is attempted 21 or more days after the deadline. If you do not think you will be able to meet this deadline please contact the editorial office immediately.

on behalf of Dr Isabelle Mareschal (Associate Editor) and Essi Viding (Subject Editor)
openscience@royalsociety.org

Associate Editor Comments to Author (Dr Isabelle Mareschal):

Associate Editor: 1

Comments to the Author:

Expert reviewers have read your manuscript and raise a number of concerns. Reviewer 1 requests a more thorough/accurate representation of the current state of the art, while reviewer 2 raises a

number of methodological issues. Please address all issues raised in the manuscript and provide a point by point reply.

Reviewer comments to Author:

Reviewer: 1

Comments to the Author(s)

The authors used a visual adaptation behavioural paradigm to test selective mechanisms to the human hand. The results are interesting, but confirm previous evidence, rather than provide novel evidence as, instead, suggested by the authors.

The experiment is sound. The main issue to me is how results are presented. Throughout the paper, the authors repeatedly suggest that their results provide novel evidence for specialized neural mechanisms selective to process the human hand. However, this evidence was long ago reported with neurophysiology in non-human primates (Desimone et al., 1984; Gross et al., 1969) and extensively with neuroimaging in humans (Bracci et al., 2010; 2013; Op de Beeck et al., 2010; Orlov et al., 2010 etc). Interestingly, an fMRI study published 10 years ago (Bracci et al., 2010) already ruled out the role of shape similarity including robotic hands among the control categories.

I would suggest toning down the writing and better acknowledge the already existing extensive evidence on this matter. For instance, the following sentences are not acceptable.

“It is still far from clear, instead, whether humans and monkeys also share specialized brain systems encoding a visual representation of hands”.

“Whether similar representation is found in humans is not known”.

The authors report some of the evidence in the literature, but then go on saying “However, no straightforward data are available on existence in humans of neural populations selectively encoding hand shape.”

The above sentences are not true and also misleading, since the authors are well aware of the existing literature. They cite the evidence but they seem to believe that “good” evidence for neural mechanisms encoding hands can be provided with behavioural data, not neuroimaging data: “Such evidence can be provided by psychophysical studies using visual adaptation”.

I would suggest presenting the results as confirmation of existing literature rather than novel data providing evidence from neural mechanisms not shown before.

Other misleading sentences that point to novelty rather than confirmatory evidence are the following:

“Our study provides evidence that the human visual system is equipped with selective neural mechanisms encoding the visual representation of human hand”.

“These results support the view that neural mechanisms representing faces can be differentiated from those representing hands, but more importantly allow to demonstrate that a shape-selective mechanism specifically coding human hand can be identified”.

Reviewer: 2

Comments to the Author(s)

The manuscript addresses the question of whether humans have neurons that selectively represent the visual shape of the primate hand, by conducting adaptation experiments using

human hands, robot hands and non-human animal paws. The authors conclude that the fact that judgements of 'human-likeness' of a visual probe representing a human hand were reduced when using a human hand adaptor but not a robot or a non-human animal one offers support for the hypothesis that there are specific neural mechanisms encoding human hands.

The stimuli section is lacking in detail: e.g. there is no information about how many different stimuli were created, and it's not clear to me what was actually morphed (I presume items from different stimulus categories were morphed e.g. in Figure 1B, the human hand looks like it's been morphed with an animal paw to me?)

I'm also a little concerned about the stimulus selection. Using only two exemplars for each category seems problematic, especially when they appear to be quite different in terms of low-level features (e.g. the paw stimuli appear to be much darker than the human hand stimuli). I appreciate that you changed the sizes of the adaptors in order to try to counteract this, but it seems to me that if the adaptors were in the same position (just larger) you might still expect low level adaptation that could transfer to the probe stimuli? I am not totally convinced that your results couldn't be explained by the differences in luminance between your stimuli and would appreciate further clarification from the authors on why they think that higher level mechanisms should be evoked.

If I understand the results on P.12 correctly, the only significant interaction effects seem to be for a small range of morphing conditions (20-35%). Some discussion of this pattern would be useful – is it what would be expected based on your stimulus design?

It would also be nice to see individual level data on Figures 3A, 4A and 5A (e.g. in the form of dots superimposed on the bar graphs) in order to get a more intuitive understanding of the variability between participants (which appears quite large, though there currently isn't any indication of what the error bars represent on these graphs).

There are a number of minor typographical errors (e.g. "we showed that participants' judgement of human-likeness of a visual probe representing (a) human hand" in the abstract on P3, incorrect font in P2 on P11, "with no differences between the four three conditions from 0-to-15% of morphing" on P12) and thus I would recommend the authors carry out a full proofread of the manuscript.

===PREPARING YOUR MANUSCRIPT===

- one version identifying all the changes that have been made (for instance, in coloured highlight, in bold text, or tracked changes);
- a 'clean' version of the new manuscript that incorporates the changes made, but does not highlight them.

This version will be used for typesetting if your manuscript is accepted.

===PREPARING YOUR REVISION IN SCHOLARONE===

-- If you have uploaded ESM files, please ensure you follow the guidance at <https://royalsociety.org/journals/authors/author-guidelines/#supplementary-material> to include a suitable title and informative caption. An example of appropriate titling and captioning

may be found at https://figshare.com/articles/Table_S2_from_Is_there_a_trade-off_between_peak_performance_and_performance_breadth_across_temperatures_for_aerobic_sc_ope_in_teleost_fishes_/3843624.

Author's Response to Decision Letter for (RSOS-200948.R0)

See Appendix A.

RSOS-200948.R1 (Revision)

Review form: Reviewer 1

Is the manuscript scientifically sound in its present form?

Yes

Are the interpretations and conclusions justified by the results?

Yes

Is the language acceptable?

Yes

Do you have any ethical concerns with this paper?

No

Have you any concerns about statistical analyses in this paper?

No

Recommendation?

Accept as is

Comments to the Author(s)

The authors addressed my comments.

Review form: Reviewer 2

Is the manuscript scientifically sound in its present form?

No

Are the interpretations and conclusions justified by the results?

Yes

Is the language acceptable?

Yes

Do you have any ethical concerns with this paper?

No

Have you any concerns about statistical analyses in this paper?

No

Recommendation?

Accept with minor revision (please list in comments)

Comments to the Author(s)

I'd like to thank the reviewers for their responses.

I have just one further question: the authors state: "As for the morphing procedure, we specify here that we did not morph together items from different categories (e.g., human hand morphed with paw) but we morphed the configural shape of each single item." I'm more familiar with adaptation paradigms where the 'morphs' are intermediate between the two categories of interest e.g. male and female face adaptation stimuli, thus creating ambiguity when a participant is asked to judge these morphs as belonging to one or the other category. However, in this case, the shape of an individual item is being morphed. Perhaps this is simply my unfamiliarity with the Photoshop terminology used, but it would be useful to explain more clearly exactly what the manipulation was designed to do - it seems that it was aimed to reduce the 'human-likeness' of the stimulus in order to create ambiguity (judging by your additional discussion of the results for the human probe/human adaptor condition). However, the paw morph stimuli in the baseline condition always seemed to have very low 'human-like' judgements - perhaps suggesting that these were fairly unambiguous stimuli and would be therefore less prone to adaptation effects? I don't think this necessarily affects the main conclusions of the paper (which predominantly seem to be based on the hand probe condition) but it would be useful to have greater clarification about these manipulations and how they might have affected the results.

Decision letter (RSOS-200948.R1)

Dear Professor Conson

On behalf of the Editors, we are pleased to inform you that your Manuscript RSOS-200948.R1 "NOT ONLY FACES": SPECIALIZED VISUAL REPRESENTATION OF HUMAN HANDS REVEALED BY ADAPTATION" has been accepted for publication in Royal Society Open Science subject to minor revision in accordance with the referees' reports. Please find the referees' comments along with any feedback from the Editors below my signature.

Please submit your revised manuscript and required files (see below) no later than 7 days from today's (ie 05-Nov-2020) date. Note: the ScholarOne system will 'lock' if submission of the revision is attempted 7 or more days after the deadline. If you do not think you will be able to meet this deadline please contact the editorial office immediately.

on behalf of Dr Isabelle Mareschal (Associate Editor) and Essi Viding (Subject Editor)
openscience@royalsociety.org

Reviewer comments to Author:

Reviewer: 1

Comments to the Author(s)

The authors addressed my comments.

Reviewer: 2

Comments to the Author(s)

I'd like to thank the reviewers for their responses.

I have just one further question: the authors state: "As for the morphing procedure, we specify here that we did not morph together items from different categories (e.g., human hand morphed with paw) but we morphed the configural shape of each single item." I'm more familiar with adaptation paradigms where the 'morphs' are intermediate between the two categories of interest e.g. male and female face adaptation stimuli, thus creating ambiguity when a participant is asked to judge these morphs as belonging to one or the other category. However, in this case, the shape of an individual item is being morphed. Perhaps this is simply my unfamiliarity with the Photoshop terminology used, but it would be useful to explain more clearly exactly what the manipulation was designed to do – it seems that it was aimed to reduce the 'human-likeness' of the stimulus in order to create ambiguity (judging by your additional discussion of the results for the human probe/human adaptor condition). However, the paw morph stimuli in the baseline condition always seemed to have very low 'human-like' judgements – perhaps suggesting that these were fairly unambiguous stimuli and would be therefore less prone to adaptation effects? I don't think this necessarily affects the main conclusions of the paper (which predominantly seem to be based on the hand probe condition) but it would be useful to have greater clarification about these manipulations and how they might have affected the results.

===PREPARING YOUR MANUSCRIPT===

===PREPARING YOUR REVISION IN SCHOLARONE===

-- Ensure that your data access statement meets the requirements at <https://royalsociety.org/journals/authors/author-guidelines/#data>. You should ensure that you cite the dataset in your reference list. If you have deposited data etc in the Dryad repository, please only include the 'For publication' link at this stage. You should remove the 'For review' link.

Author's Response to Decision Letter for (RSOS-200948.R1)

See Appendix B.

Decision letter (RSOS-200948.R2)

Dear Professor Conson,

It is a pleasure to accept your manuscript entitled "'NOT ONLY FACES": SPECIALIZED VISUAL REPRESENTATION OF HUMAN HANDS REVEALED BY ADAPTATION" in its current form for publication in Royal Society Open Science.

on behalf of Dr Isabelle Mareschal (Associate Editor) and Essi Viding (Subject Editor)
openscience@royalsociety.org

Appendix A

To: Dr Isabelle Mareschal
Associate Editor
Royal Society Open Science

Dear Editor,

We are submitting the revised version of the paper RSOS-200948: "NOT ONLY FACES": SPECIALIZED VISUAL REPRESENTATION OF HUMAN HANDS REVEALED BY ADAPTATION. In revising the text, we took into account all the comments raised by the Reviewers. We thank for the comments that helped us to strengthen the manuscript and hope that it is now suitable for publication in RSOS.

Kind regards
Massimiliano Conson

Reviewer comments to Author:

Reviewer: 1

Point 1. The authors used a visual adaptation behavioural paradigm to test selective mechanisms to the human hand. The results are interesting, but confirm previous evidence, rather than provide novel evidence as, instead, suggested by the authors.

The experiment is sound. The main issue to me is how results are presented. Throughout the paper, the authors repeatedly suggest that their results provide novel evidence for specialized neural mechanisms selective to process the human hand. However, this evidence was long ago reported with neurophysiology in non-human primates (Desimone et al., 1984; Gross et al., 1969) and extensively with neuroimaging in humans (Bracci et al., 2010; 2013; Op de Beeck et al., 2010; Orlov et al., 2010 etc). Interestingly, an fMRI study published 10 years ago (Bracci et al., 2010) already ruled out the role of shape similarity including robotic hands among the control categories.

I would suggest toning down the writing and better acknowledge the already existing extensive evidence on this matter. For instance, the following sentences are not acceptable:

"It is still far from clear, instead, whether humans and monkeys also share specialized brain systems encoding a visual representation of hands".

"Whether similar representation is found in humans is not known".

The authors report some of the evidence in the literature, but then go on saying "However, no straightforward data are available on existence in humans of neural populations selectively encoding hand shape."

The above sentences are not true and also misleading, since the authors are well aware of the existing literature. They cite the evidence but they seem to believe that "good" evidence for neural mechanisms encoding hands can be provided with behavioural data, not neuroimaging data: "Such evidence can be provided by psychophysical studies using visual adaptation".

I would suggest presenting the results as confirmation of existing literature rather than novel data providing evidence from neural mechanisms not shown before.

Other misleading sentences that point to novelty rather than confirmatory evidence are the following:

“Our study provides evidence that the human visual system is equipped with selective neural mechanisms encoding the visual representation of human hand”.

“These results support the view that neural mechanisms representing faces can be differentiated from those representing hands, but more importantly allow to demonstrate that a shape-selective mechanism specifically coding human hand can be identified”.

Response. *We thank the Reviewer for the comments. We are sorry for having given the impressions that “good” evidence for neural mechanisms encoding hands can be provided with behavioural rather than with neuroimaging data. We aimed at providing evidence for existing neural specificity of hand representation by the technique of visual adaptation. In this respect, our data are novel, whereas as correctly underscored by the Reviewer, we could provide support to evidence from available neuroimaging literature on the existence of a human hand representation in human brain. Indeed, the present results from visual adaptation technique could be nicely compared with data from classical neurophysiological studies in non-human primates (Desimone et al., 1984; Gross et al., 1969), since psychophysical adaptation experiments are used to infer the tuning properties of cells underlying the perception of specific stimuli, due to the fact that the responses of neurons tuned to the adapting stimulus are selectively reduced by repeated exposure (Kohn, 2007 for a review). Following these considerations, we toned down the writing and better acknowledged the already existing neuroimaging evidence on hand representation in humans. In particular, we paid specific attention to all the sentences underscored by the Reviewer, and also carefully checked the text to find further sentences possibly suffering from the above misconception.*

Reviewer: 2

Point 1. The stimuli section is lacking in detail: e.g. there is no information about how many different stimuli were created, and it’s not clear to me what was actually morphed (I presume items from different stimulus categories were morphed e.g. in Figure 1B, the human hand looks like it’s been morphed with an animal paw to me?).

Response. *In the revised manuscript we provided further details about stimuli, by clarifying that all images were acquired in full colour, then, they were converted to greyscale and matched for contrast and brightness by using Adobe Photoshop CS5 software. A morphing procedure was thus applied through the “scale based on content” function, leaving the focus of the process unchanged for all images (Adobe Photoshop CS5 software). As for the morphing procedure, we specify here that we did not morph together items from different categories (e.g., human hand morphed with paw) but we morphed the configural shape of each single item.*

Point 2. I’m also a little concerned about the stimulus selection. Using only two exemplars for each category seems problematic, especially when they appear to be quite different in terms of low-level features (e.g. the paw stimuli appear to be much darker than the human hand stimuli). I appreciate that you changed the sizes of the adaptors in order to try to counteract this, but it seems to me that if the adaptors were in the same position (just larger) you might still expect low

level adaptation that could transfer to the probe stimuli? I am not totally convinced that your results couldn't be explained by the differences in luminance between your stimuli and would appreciate further clarification from the authors on why they think that higher level mechanisms should be evoked.

Response. *We thank the Reviewer for the comments. In the original text we did not specify in the methods section that all images were converted to greyscale and matched for contrast, brightness using Adobe Photoshop, in order to control low-level differences between stimuli potentially affecting the adaptation procedure. Moreover, as also underscored by the Reviewer, test images were presented smaller than adapting images, to minimize any contribution from low-level retinotopic after-effects, as done in previous studies (Zhao and Chubb, 2001; Jenkins et al., 2006; Jeffery et al., 2006; Anderson and Wilson, 2005; Guo et al., 2009; Oruç and Barton, 2010; Lawson 2009). Adaptation to low-level retinotopic properties seems unlikely because subjects were not required to stabilize their gaze during the adaptation period of 6 sec (without fixation point). Anyway, even in conditions where a stabilization of the participants' gaze is required during the adaptation period, the test stimuli we used here were considerably different in size from the adapting stimuli. Such a methodological caution allows to avoid low-level features adaptation, as previously underscored. Another interesting result of our study is that we find a category-specific aftereffect where a hierarchical hand representation could be envisaged, with a higher and canonical representation of human hand combining both function and morphology, and a lower human hand representation only defined by the functional specificity of the anthropoid hand, with no adaptation for a different specimen probe. This hierarchical representation of the aftereffect leads our conclusion that a high-level mechanism should be involved in generating these effects. However, further studies with specific different low-level modulation (e.g. orientation, contrast viewpoint) are warranted to shed light on the source of the phenomenon we could demonstrate here.*

Point 3. If I understand the results on P.12 correctly, the only significant interaction effects seem to be for a small range of morphing conditions (20-35%). Some discussion of this pattern would be useful – is it what would be expected based on your stimulus design?

Response. *Our results show that in the hand condition, the hand adaptor significantly affected human-likeness of the hand probe at levels of morphing representing the centre of the morphing space, i.e., 20%-40%. This pattern of adaptation could be explained by the fact that at the boundaries of the morphing range the stimuli were non-ambiguous (human-likeness values were higher at lower morphing levels and, instead, were lower at higher morphing levels) while in the 20%-40% range the hand stimuli appeared more ambiguous (human-likeness values were placed in the middle range of the human-likeness scale). This interpretation is consistent with available literature showing that aftereffects are minimal for non-ambiguous stimuli, while they are stronger for more ambiguous stimuli (Kessler et al. 2013). We reported this comment in the Discussion section.*

Point 4. It would also be nice to see individual level data on Figures 3A, 4A and 5A (e.g. in the form of dots superimposed on the bar graphs) in order to get a more intuitive understanding of the variability between participants (which appears quite large, though there currently isn't any indication of what the error bars represent on these graphs).

Response. We thank the referee to highlight this lack of clarity. Now we added a description in the caption of the figures that states what error bars represent (1 SEM). Moreover, we added supplementary figures with box plots displaying individual data for all experimental conditions.

Point 5. There are a number of minor typographical errors (e.g. “we showed that participants’ judgement of human-likeness of a visual probe representing (a) human hand” in the abstract on P3, incorrect font in P2 on P11, “with no differences between the four three conditions from 0-to-15% of morphing” on P12) and thus I would recommend the authors carry out a full proofread of the manuscript.

Response. We thank the Reviewer for having underscored these errors that we amended in the revised text. We also carried out a full proofread of the manuscript.

Appendix B

To: Dr Isabelle Mareschal
Associate Editor
Royal Society Open Science

Dear Editor,

We are happy that the manuscript has been accepted for publication in Royal Society Open Science after minor revision. We are now submitting the reviewed version taking into account the comment of Reviewer #2. We thank for this further comment that helped us to make clearer one more methodological aspect of the study.

Kind regards
Massimiliano Conson

Reviewer: 2

Comments to the Author(s)

Point 1. I have just one further question: the authors state: “As for the morphing procedure, we specify here that we did not morph together items from different categories (e.g., human hand morphed with paw) but we morphed the configural shape of each single item.” I’m more familiar with adaptation paradigms where the ‘morphs’ are intermediate between the two categories of interest e.g. male and female face adaptation stimuli, thus creating ambiguity when a participant is asked to judge these morphs as belonging to one or the other category. However, in this case, the shape of an individual item is being morphed. Perhaps this is simply my unfamiliarity with the Photoshop terminology used, but it would be useful to explain more clearly exactly what the manipulation was designed to do – it seems that it was aimed to reduce the ‘human-likeness’ of the stimulus in order to create ambiguity (judging by your additional discussion of the results for the human probe/human adaptor condition).

However, the paw morph stimuli in the baseline condition always seemed to have very low ‘human-like’ judgements – perhaps suggesting that these were fairly unambiguous stimuli and would be therefore less prone to adaptation effects? I don’t think this necessarily affects the main conclusions of the paper (which predominantly seem to be based on the hand probe condition) but it would be useful to have greater clarification about these manipulations and how they might have affected the results.

Response. We agree with the Reviewer that the procedure used in the present experiment was different from the typical adaptation paradigm usually employed for faces or objects. Indeed, here, we did not morph together items from different categories, but we manipulated the configural shape of the stimuli using the scale based on content of Adobe Photoshop CS5 software. In the revised text, we further specified as follows: “A morphing procedure was thus applied through the “scale based on content” function of Adobe Photoshop CS5 software, modifying each image on both the horizontal and the vertical dimensions and varying the value of

the 5% function at regular intervals from 0% to 60%. By this means, we could manipulate the human-likeness of the images by modifying the configural shape of the hand (or paw) while leaving the focus of the process unchanged for all images (Figure 1B)". Importantly, to verify whether human-likeness of the images was actually manipulated and whether human-likeness of two items within each category was comparable and differed from human-likeness of each item of the other categories, we performed a pilot study that confirmed that our morphing procedure manipulated human-likeness of the stimuli as expected. Relevantly, moreover, as far as paw stimulus is concerned, results of the pilot study confirmed the authors' intention of providing participants with an easier judgment with respect of human-likeness, as they were presented with a stimulus category that was less ambiguous, as clearly not human. Results of the main experiment confirmed that since the paw was less ambiguous it was less susceptible to visual adaptation. Indeed, data showed that there was no difference of human-likeness judgment between the different levels of morphing. Therefore, we could consider the paw condition as a control condition allowing us to show that adaptation worked well when the stimulus to adapt was ambiguous with respect to human-likeness and not when it clearly belonged to a category as the animal one. Indeed, as we reported in the discussion section, the main data demonstrated that human-hand adaptation was in play for more ambiguous stimuli (Hand and Robot), whereas no effect was observed when using non-primate animal paw as either adaptor or probe.